# Physiological Analysis and Transcriptome Profiling of Inverted Cuttings of *Populus yunnanensis* Reveal That Cell Wall Metabolism Plays a Crucial Role in Responding to Inversion

**DOI:** 10.3390/genes9120572

**Published:** 2018-11-23

**Authors:** An-Pei Zhou, Dan Zong, Pei-Hua Gan, Xin-Lian Zou, Xuan Fei, Yuan-Yuan Zhong, Cheng-Zhong He

**Affiliations:** 1Key Laboratory for Forest Genetic and Tree Improvement and Propagation in Universities of Yunnan Province, Southwest Forestry University, Kunming 650224, China; zhouanpei85@sina.cn (A.-P.Z.); zdkathy@163.com (D.Z.); ganpeihua33@sina.cn (P.-H.G.); 13529154758@163.com (X.-L.Z.); feixuan0101@163.com (X.F.); zhongyuanyuan221@sina.com (Y.-Y.Z.); 2Key Laboratory of Biodiversity Conservation in Southwest China, State Forestry Administration, Southwest Forestry University, Kunming 650224, China; 3Key Laboratory for Forest Resources Conservation and Utilization in the Southwest Mountains of China, Ministry of Education, Southwest Forestry University, Kunming 650224, China

**Keywords:** *Populus yunnanensis*, inversion, cell wall, endogenous hormone, oxidase, transcriptome profiling

## Abstract

Inverted cuttings of *Populus yunnanensis* remain alive by rooting from the original morphological apex and sprouting from the base, but the lateral branches exhibit less vigorous growth than those of the upright plant. In this study, we examined the changes in hormone contents, oxidase activities, and transcriptome profiles between upright and inverted cuttings of *P. yunnanensis*. The results showed that the indole-3-acetic acid (IAA) and gibberellic acid (GA_3_) contents were significantly lower in inverted cuttings than in upright cuttings only in the late growth period (September and October), while the abscisic acid (ABA) level was always similar between the two direction types. The biosynthesis of these hormones was surprisingly unrelated to the inversion of *P. yunnanensis* during the vegetative growth stage (July and August). Increased levels of peroxidases (PODs) encoded by 13 differentially expressed genes (DEGs) served as lignification promoters that protected plants against oxidative stress. Kyoto encyclopedia of genes and genomes (KEGG) enrichment analysis showed that most DEGs (107) were related to carbohydrate metabolism. Furthermore, altered activities of uridine diphosphate (UDP)-sugar pyrophosphorylase (USP, 15 DEGs) for nucleotide sugars, pectin methylesterase (PME, 7 DEGs) for pectin, and POD (13 DEGs) for lignin were important factors in the response of the trees to inversion, and these enzymes are all involved cell wall metabolism.

## 1. Introduction

*Populus yunnanensis*, a native dioecious poplar species of southwest China, is widely distributed in low-latitude and high-elevation areas [1]. *P. yunnanensis* is a dominant tree in forestry production and environmental protection due to its fast growth and easy cutting propagation [2]. After cutting, we accidentally found that inverted cuttings of *P. yunnanensis* survived and could form a complete tree. First, rooting at the original morphological apex and sprouting at the base ensured that the cutting remained alive. Then, the inverted cuttings showed deferred sprouting and bending-up growth. Finally, the lateral branch that sprouted from the base bud of the stem exhibited strong growth, although it was less vigorous than that of the upright branch. This growth response to plant inversion is of interest.

Polarity, an essential characteristic of differentiation along an axis of symmetry, designates the specific orientation of activity in space [3,4]. This concept involves no assumptions whatsoever regarding its causes and has been used to describe (1) growth gradients, (2) the special transport direction of auxin, (3) the spatial order of cell divisions, and (4) structural or physiological differences between two ends of a cell [5]. Inversion of plants was reported in the twentieth century [5,6,7,8,9,10]. Early experiments in plants focused on what factors oriented polarity and whether the original polarity could be reversed by transduction of external stimuli, such as gravity and light [5,7,8,10,11,12]. Tow systems are involved in this phenomenon: thallus-rhizoid polarity in phaeophycean and pteridophytean plant and shoot-root polarity in higher plants [5]. In phaeophycean zygotes and bryophytean and pteridophytean spores, an unpolarized state is observed initially and can be redistributed by external stimuli. In flowering plants, shoot-root polarity has been studied extensively in various species. Considering observations on morphological induction and hormone gradients [5,13,14,15], major investigators hypothesized that polarity that is detected in the zygote can be latent but cannot be reversed.

In the 21st century, a number of studies focused on how polarity is induced and fixed in plant embryogenesis and how it orients cell division [16,17,18,19,20,21], although few studies have examined the growth and development of inverted plants. Plants are characterized by a high degree of flexibility, and can alter their morphogenesis to respond to the environment [22,23]. There are various factors influencing this process, in which hormone regulation plays a critical role. Nine principal plant hormones (auxin indole-3-acetic acid, IAA), gibberellins (GAs), cytokinins (CTKs), abscisic acid (ABA), ethylene (ET), brassinosteroids (BRs), jasmonic acid (JA), salicylic acid (SA), and strigolactones (SLs) are linked to growth regulation [24,25,26,27,28,29]. Previous studies showed that auxin, GA, BR, SL, and CTK are related to normal plant growth and development [23,30,31], while ABA, ET, JA, and SA mainly contribute to the responses to various stresses [32,33,34]. Among these hormones, auxin is recognized as a plant hormone that regulates cell proliferation and promotes cell elongation through gradients, and GA is responsible for stem extension based on cell elongation [23,35]. Notably, these hormones can function independently, but their interactions are the main regulators in many processes. Multiple hormones are involved in plant growth regulation and have overlapping influences on various cellular processes. For example, GA can mediate many processes, including seed germination, hypocotyl elongation, root growth, lateral organ formation, flowering, leaf serration, internode elongation, and reproduction, and these processes are also controlled by auxin [22,24,36,37].

The growth and development of plants are closely related to the environment. Abiotic stresses commonly block plant growth, as shown by reduced crop yields under cold and salt stress [38,39] and the development of dwarf Polish wheat under metal stress [40]. Stress alters the intrinsic metabolic balance of the plant body. For example, excessive production of reactive oxygen species (ROS) can occur. These species, such as H_2_O_2_, are highly reactive and seriously disrupt normal metabolism through damage to lipid membranes, proteins, and nucleic acids [41,42,43]. Peroxidase (POD) is highly involved in mediating the concentration of H_2_O_2_ [44]. PODs are primarily H_2_O_2_-reducing enzymes and can also promote the formation of H_2_O_2_, which is required for lignification [45]. Polyphenol oxidase (PPO) is an enzyme that catalyzes the formation of o-quinones [46]. Due to its possible involvement in plant defense, PPO shows changes in activity induced by biotic and abiotic stresses [47]. IAA oxidase (IAAO) plays an important role in the enzyme defense of plants via a regulator of IAA content [48]. The activities of these three oxidases were assessed in relation to the elevation of responses to stress, both alone and in combination. Moreover, ABA is a stress hormone that is sensitive to environmental cues and can induce many cellular mechanisms associated with stress resistance [49,50].

The control of developmental aspects and growth is a complex biological process with major significance for morphology. High-throughput RNA sequencing (RNA-Seq) is an attractive approach for transcriptome profiling and a powerful tool for examining this process [51,52,53]. RNA-Seq can provide precise measurements of transcripts and does not require prior gene information, as in other techniques such as microarray analyses [54,55]. Transcripts were annotated in nucleotide and protein databases such as the non-redundant nucleotide sequences (NT), clusters of orthologous groups of proteins (COG), and Kyoto encyclopedia of genes and genomes (KEGG). Statistical analyses such as Fisher’s exact test were used to enrich differentially expressed genes (DEGs) in various classes of the databases. In this study, we characterized the growth of upright and inverted cuttings of *P. yunnanensis* and compared their hormone contents, oxidase activities, and transcriptome profiles. The goal was to reveal the roles of hormones and oxidase in response to the inversion and the most important factors impacting the growth of inverted plants.

## 2. Materials and Methods

### 2.1. Plant Materials

The main stems of one-year-old *P. yunnanensis* clones were chosen to produce cuttings. One clone was treated as a biological replicate and three replicates were established. At a screening size of approximately 80 cm in length and 10 mm in diameter, cuttings from every clone were cultured in greenhouses of Southwest Forestry University (Kunming, China) in two direction types: upright and inversion (Appendix A). Red soil, a common soil type in southwest China, was used as the medium for cutting propagation. Without supplemental light, the conditions of the greenhouse were customized, with regular temperatures of approximately 28 °C during the day and 20 °C at night. The experiment was established in early March, and the cuttings were watered every 2 or 3 d. No prophylactic procedures for disease or pests were applied.

We removed upper and lower buds to ensure the growth of the middle bud of the cuttings. In early April, the cuttings showed sprouting from the middle bud and the branch was established. All of the top leaves (including the top shoot) of the branch were sampled in July, August, September, and October, where July and August were considered to represent the vegetative growth stage, and September and October were considered to represent the late growth stage. The top leaves of four individuals from both clone and direction types were pooled to generate one mixed leaf sample, and six mixed samples from the two direction types, with three biological replicates, were collected in every month. In total, twenty-four leaf samples from the four months were used to compare the differences in hormones and oxidases between upright and inverted cuttings. The six samples collected only in August (vegetative growth stage) were used to characterize the transcriptome profile. All of the leaf materials were frozen in liquid nitrogen and stored at −80 °C.

### 2.2. Growth Comparison

To detect differences in growth between upright and inverted cuttings, we measured the width and length of the main branches. The results from four individuals belonging to one mixed sample were averaged to obtain the mean value, and an independent samples *t*-test was performed using R software [56].

### 2.3. Extraction of Endogenous Hormones and Quantification

We employed high-performance liquid chromatography (HPLC) to determine the concentrations of IAA, ABA, and gibberellic acid (GA_3_). Leaf samples of more than 0.5 g were first weighed and then ground in 10 mL of 99.8% methanol for 24 h at 4 °C. The samples were next purified with 50 mL of petroleum ether. After evaporation in a rotary evaporator under vacuum at 40 °C, purification of IAA, ABA, and GA_3_ was performed using ethyl acetate. Finally, the residues were dissolved in 1 mL of 99.8% methanol, and the supernatant was filtered through a 0.45 μm-pore size filter for measurement. HPLC analysis was carried out using an Agilent C_18_ column (Agilent Technologies, Palo Alto, CA, USA) maintained at 30 °C with a flow rate of 1 mL·min^−1^. The hormone concentrations were determined at 254 nm for IAA and ABA and at 210 nm for GA_3_ and were subjected to statistical analysis.

### 2.4. Extraction of IAAO, PPO, and POD

Assays for IAAO, PPO, and POD were performed as described by Kar and Mishra [57], Beffa et al. [58], and Rout [59]. In this step, approximately 0.1 g of leaves were ground to a homogenate with 1 mL of 0.05 mol·L^−1^ sodium phosphate in an ice bath followed by centrifugation for 20 min at 12,000 *g* at 4 °C. The supernatant was used for determination of oxidase activities. For IAAO analysis, a reaction solution containing 1 mL of MnCl_2_, 2 mL of IAA, 1 mL of crude enzyme extract, and 5 mL of phosphate buffer was incubated for 30 min at 30 °C in darkness. Controls were obtained by adding phosphate buffer to replace the crude enzyme extract and adding distilled water to replace IAA. The absorbance at 530 nm was measured to determine residual IAA, and a standard curve was constructed based on ten IAA concentrations (0, 0.5, 1, 2.5, 5, 10, 15, 20, 25, and 35 μg·mL^−1^). PPO activity was determined in a reaction solution including 0.2 mL of 0.3% H_2_O_2_, 1 mL of 0.1% guaiacol, 0.1 mL of crude enzyme extract, and 2 mL of phosphate buffer. The absorbance at 530 nm was measured to determine the change in the value every 15 s. POD activity was determined in a reaction solution including 1 mL of 0.1 mol·L^−1^ H_2_O_2_, 0.3 mL of crude enzyme extract, and 2 mL of phosphate buffer. The absorbance at 470 nm was measured to determine the change in the value every 15 s. We compared the activities of the three enzymes and assessed their statistical significance with an independent samples *t*-test.

### 2.5. RNA Isolation and Library Construction

Total RNA from the top leaf samples of upright (LU) and inverted (LI) cuttings of *P. yunnanensis* was isolated using the RNAprep Pure Plant Kit (Tiangen Biotech, Beijing, China). The RNA concentration was determined using a spectrophotometer at A260/280 (NanoDrop 1000, Thermo Scientific, Wilmington, DE, USA), and the integrity was assessed in an Agilent 2100 system (Agilent Technologies). The qualified RNA samples were used for complementary DNA (cDNA) library construction. Paired-end reads were obtained based on the Illumina HiSeq 4000 platform at the Beijing Genomics Institute (BGI) Co. Ltd. (Beijing, China), and each library produced approximately 6 gigabases (Gb) of raw data. The transcriptome data in this study were submitted to the short read archive (SRA) database under the accession number PRJNA505895.

### 2.6. De Novo Assembly and Functional Annotation

For quality control, the raw reads were trimmed and filtered to remove Illumina adapter sequences, low-quality reads, and reads containing poly-N sequences (>5%) using in-house Perl scripts. The processed clean reads were assembled de novo using Trinity software [60], and the obtained unigenes were subjected to basic local alignment search tool (BLAST) searched [61] against the non-redundant protein sequences (NR), NT, SwissProt, KEGG, COG, and Interpro databases. With NR annotation, Blast2GO [62] was applied to obtain gene ontology (GO) annotations of unigenes.

### 2.7. Sample Correlation and Differentially Expressed Gene Detection

Unigenes from all six samples were clustered, and an all-unigene group was generated as a reference transcriptome. The clean reads of the samples were then mapped back to the reference library using Bowtie2 software [63]. In this step, the numbers of reads mapped back to each unigene were counted. We calculated the fragments per kilobase of transcript per million mapped reads (FPKM) to assess expression levels using RSEM software [64] and chose those with an FPKM ≥ 0.5 for downstream analysis. We detected the DEGs between the upright and inverted groups using the NOIseq method [65] and chose those with a fold change ≥2 and probability ≥0.8. TBtools [66] was used for the significant enrichment of DEGs among GO terms and KEGG pathways, and their visualization was carried out in the ggplot2 package of R software [67].

### 2.8. Real-Time Quantitative PCR Analysis

To confirm the RNA-seq results, we chose ten DEGs from the pathway analysis and quantified them using real-time quantitative PCR (RT-qPCR). In this step, leaf total RNA was extracted with three biological replicates as described above. The gene-specific primers were designed using Primer Premier 5 [68], and their characteristics are listed in Appendix A. PED1, one endogenous control gene, was selected as an internal standard. Two-step amplification with three technical replicates was performed using Fast Super EvaGreen qPCR Master Mix (US Everbright Inc., Suzhou, China) on a Rotor-Gene Q real-time PCR system (Qiagen, Germany). We calculated relative expression levels with the 2^−∆∆Ct^ method [69].

## 3. Results

### 3.1. Effect of Inversion on Growth, Hormones, and Oxidases in Cuttings

The width and length of the main branch of the upright cuttings were greater than those of the inverted cuttings, and these differences were significant at the 0.01 level (Figure 1A). During the entire observation period, the branch growth of upright cuttings was obviously improved, with maximal increases (6.27 mm for branch width; 71.83 cm for branch length) being observed from July to August, while inverted cuttings showed slow growth, with maximal increases (0.91 mm for branch width; 5.73 cm for branch length) also occurring from July to August.

The variance analysis showed that the ABA contents were similar between the two direction types (Figure 1B). The inverted cuttings exhibited significantly lower levels of IAA and GA_3_ than the upright cuttings in September and October. In August, the GA_3_ content of inverted cuttings was as high at 1519.76 μg·g^−1^·FW, but was not significantly different from that of the upright cuttings. The coefficients of correlation between the three endogenous hormones ranged from 0.552 (ABA vs. IAA, upright) to 0.931 (GA_3_ vs. IAA, inversion), and they were all significant. We found that the correlations of the hormones in inverted cuttings were always higher than those in upright cuttings.

As shown in Figure 1C, the inverted cuttings exhibited lower IAAO activity than the upright cuttings in July, September, and October, and the opposite pattern was observed in August. All comparisons were significant at the 0.05 level. PPO and POD exhibited similar patterns between the two direction types, and the activity values of inverted samples were higher than those of upright samples in all observation periods. These differences were significant except for the change in PPO in September. The correlations between the three enzymes were high, ranging from 0.431 (IAAO vs. PPO, inversion) to 0.906 (IAAO vs. POD, upright), and they were all significant. We noted that the Pearson coefficients of IAAO vs. PPO and IAAO vs. POD were obviously reduced when *P. yunnanensis* cuttings were inverted.

Taking upright cuttings as a control, we observed a change in the correlations between growth, hormones, and enzymes (Figure 2). The branch length of inverted cuttings exhibited no significant correlations with GA_3_, IAA, PPO, and POD, and the GA_3_ content was not significantly related to PPO and POD. The coefficient of correlation between IAA and PPO decreased from 0.52 (significant) to 0.07 (not significant), but that between IAA and IAAO increased from 0.25 (not significant) to 0.72 (significant).

### 3.2. Transcriptome Sequencing, De Novo Assembly, and Functional Annotation

The RNA-Seq libraries generated an average of 64.24 Mb raw reads (Table 1). After quality control, an average value of 44.71 Mb for clean reads was obtained from six samples. With a Q20 > 97.86% and Q30 > 93.9%, more than 68 thousand unigenes per sample were assembled, and their N50 and N90 values were greater than 801 and 238, respectively. In total, 296,815 unigenes were obtained from all unigenes, and they were annotated to seven databases (Figure 3). More than 85.29% of the unigenes exhibited successful hits in the NT database, and the ranges were 45.37~70.00% in the KEGG database and 41.96~66.66% in the GO database.

### 3.3. DEG Detection between Upright and Inverted Samples

We compared the changes in unigene expression in *P. yunnanensis* samples grown in different directions and found an obvious difference. There were 6512 DEGs in the LU vs. LI pair, suggesting a major transcriptome change induced by inversion. Comparisons of the two direction types (Figure 4) identified a few common DEGs (45) and fewer unique DEGs were detected in group LU (1511) than in group LI (4956). We also observed that the number of upregulated unigenes (4995) was higher than that of downregulated unigenes (1517) (Figure 5A), which indicated an increased transcription function in response to inversion.

The GO enrichment analysis (Figure 5B) showed that 640 DEGs could be annotated, and they were categorized into three main classifications. The biological process category exhibited 19 GO terms, among which the metabolic process was the most abundant term, followed by the cellular process. Of the 16 GO terms mapped to cellular components, the enriched terms were cell, cell part, membrane, membrane part, and organelle. In the molecular function ontology, nine GO terms were identified. Among these terms, binding and catalytic activities were dominant.

The KEGG database was used to determine the enrichment of all DEGs, as shown in Figure 5C,D. Most DEGs (317) were associated with metabolic pathways, among which carbohydrate metabolism showed the largest transcriptome change (107) induced by inversion. Furthermore, the LU vs. LI pair showed significant differences in the following seven pathways: amino sugar and nucleotide sugar metabolism, pentose and glucuronate interconversions, galactose metabolism, phenylpropanoid biosynthesis, phosphatidylinositol signaling system, ascorbate and aldarate metabolism, and cyanoamino acid metabolism. Plant-pathogen interactions exhibited the largest number of DEGs, but they were not significantly enriched pathways. To visualize the pathways, we added all DEGs annotated in the KEGG database into an extensive metabolism map using iPath3 web-based tools (Figure 6). Overall, 848 DEGs identified by 521 KEGG orthology identities (KO IDs) were mainly mapped to pathways such as metabolism of carbohydrates, energy, nucleotides, cofactors and vitamins, lipids, and other amino acids. Pathways including nucleotide sugar biosynthesis and glucuronate metabolism, ascorbate biosynthesis, beta-oxidation, and pectin degradation showed enhancement, which was similar to the results of KEGG enrichment analysis.

We examined the transcriptome profiles associated with hormones and oxidases, as shown in Figure 7A–E. Unigene130397_All was upregulated with a log2FoldChang of 7.67; this gene encodes aldehyde dehydrogenase (NAD+), which catalyzes the conversion of indole-3-acetaldehyde to indole-3-acetic acid. Two DEGs, Unigene12998_all and Unigene12988_all, were annotated as CYP83B and UGT74B, respectively, and both markedly impacted glucobrassicin activation. The enzyme 9-cis-epoxycarotenoid dioxygenase (NCED) is a key oxidase in ABA synthesis. Only CL8511.Contig6_All was upregulated with a log2FoldChang of 4.79. For GA, CL13722.Contig1_All and CL13722.Contig2_All, which both encode ent-copalyl diphosphate synthase (CPS), were differentially expressed. The sequences of 14 unigenes were identical to partial sequences of PPO, and their expression levels were similar. Of the 324 unigenes involved in the catalytic reaction under POD, 13 were identified as DEGs.

### 3.4. Real-Time Quantitative PCR Analysis

To test the reliability of the RNA-Seq results, we chose 10 unigenes involved in 13 pathways (Appendix A) that showed large differences in expression between LU and LI for verification via RT-qPCR. They all corresponded to the expression regulator, as shown by the log2FoldChang (Appendix A).

## 4. Discussion

### 4.1. Role of Hormones in Response to the Inversion of Plants

In the present study, we compared the hormone contents of IAA, ABA, and GA_3_ between upright and inverted cuttings of *P. yunnanensis* and further characterized their dynamic changes in response to inversion. Physiological analysis showed that ABA alterations were not related to inversion. The contents of IAA and GA_3_ were significantly lower in inverted samples than in upright samples only in September and October, suggesting a similarity in the fast growth period (July and August) and a difference in the late growth period (September and October).

ABA responds to environmental alterations such as drought regulation by changing the leaf stomatal aperture [70]. ABA belongs to a class of isoprenoids (terpenoid), and its biosynthesis involves the production of many intermediates (Figure 7B). In plastids, mevalonate is first catalyzed to form zeaxanthin, an oxygenated carotenoid, via a series of enzymatic reactions with intermediate products including isopentenyl pyrophosphate (IPP), farnesyl pyrophosphate (FPP), phytoene, lycopene, and β-carotene. Zeaxanthin epoxidase (ZEP) next promotes the synthesis reaction from zeaxanthin to violaxanthin [71,72]. Violaxanthin is then converted to 9-cis-violaxanthin and 9-cis-neoxanthin, which are substrates of the cleavage reaction of xanthoxin [73]. The enzyme 9-cis-epoxycarotenoid dioxygenase (NCED) plays a key role in this cleavage process [73,74]. When xanthoxin migrates to the cytosol, the catalytic synthesis of ABA is finally mediated through three possible pathways: abscisic aldehyde, abscisic alcohol, and xanthoxic acid pathways [75]. The ABA content is increased via ABA biosynthesis in adverse environmental conditions [49,76]. According to signal transduction, increased ABA acts in cellular responses to salt, heat, drought, and cold stress [49,77,78,79]. Previous studies on ABA also confirmed the essential role of ABA in plant defense against abiotic stress [34]. However, the transcriptome data used in this study showed that only CL8511.Contig6_All, which was homologous to NCED, was clearly upregulated in inverted *P. yunnanensis* cuttings, with a log2FoldChange value of 4.79. This change led to the greatest xanthoxin production observed and a possible increase in ABA content. Indeed, ABA levels were higher in inverted (128.92 μg·g^−1^ FW) than in upright (127.50 μg·g^−1^ FW) samples according to HPLC detection, but this difference was small and was below the level of significance. We therefore concluded that the biosynthesis and accumulation of ABA in leaves are unrelated to the inversion of cuttings, although ABA plays important roles under abiotic and biotic stresses.

IAA and GA can promote vegetative growth and oppose the growth-inhibiting property of ABA [23,27,32,34]. The accumulation of these hormones is correlated with a shift from growth to adaption to adverse situations [34,37]. In the biosynthetic pathway of GA (Figure 7B), two differentially expressed unigenes (upregulated CL13722.Contig6_All and downregulated CL13822.Contig2_All) were both annotated to the ent-copalyl diphosphate/ent-kaurene synthase gene (CPS-KS) enzyme in the catalytic synthesis of ent-copalyl pyrophosphate with geranylgeranyl pyrophosphate and led to a final increase in the GA_3_ level in inverted *P. yunnanensis* cuttings, as revealed via HPLC. Similar to the findings for ABA, this increase was small, and the difference between upright and inversion conditions was insignificant.

For IAA biosynthesis, there are two routes based on whether tryptophan (Try) is required: the Try-dependent route and the Try-independent route [80,81]. The Try-dependent route can be divided into at least four pathways (Figure 7A): the indole-3-pyruvate (IPA), tryptamine (Tam), indole-3-acetonitrile (IAN), and indole-3-acetamide (IAM) pathways. Conversion with IPA or IAM is considered the main auxin biosynthesis pathway [82,83]. Three unigenes were differentially expressed between the upright and inverted samples, among which the upregulated Unigene130397_All was annotated as aldehyde dehydrogenase (ALDH). This increasing oxidase activity could promote the catalytic synthesis of indole-3-acetaldehyde (IAAld) to IAA, but the content of IAA in inverted samples was reduced non-significantly based on HPLC detection. This finding suggests that enzyme alteration in the Tam pathway contributes little to the final content of IAA. Auxin biosynthesis is extremely complex, and multiple pathways are involved. The potential pathways should to be confirmed through further analysis and elucidation of the enzymes and genes involved in IAA biosynthesis.

### 4.2. Role of Oxidases in Response to Inversion of Plants

The three oxidases examined in this study showed obvious differences between upright and inverted cuttings of *P. yunnanensis*. Inversion impacted IAAO activity, with the values observed under inversion being higher than those of upright samples only in August. The activities of PPO and POD in inverted *P. yunnanensis* were always higher than those in upright samples in the four examined periods, and their differences were significant at the 0.05 level.

Plant PPO plays a recognized role in browning the wounding produced by pathogen and arthropod attacks. PPO activity impacts the catalytic reaction from monophenols and/or o-diphenols to o-diquinones (Figure 7D), which results in the generation of ROSs and protein complexes (i.e., brown melanin pigments) [84]. PPO has also been extensively investigated for its possible involvement in plant defense [47,84], and its activity is clearly related to the response to environmental changes under water stress and salt stress [85,86]. In this study, PPO activity was increased upon exposure to inverted conditions, and the difference was greatest [1158.85 U·(μg·FW·min)^−1^] in October, which suggests a high impact during the late growth period. This enzyme is listed as EC 1.10.3.1 in the KEGG database, based on the nomenclature of the International Union of Biochemistry and Molecular Biology (IUBMB), and it is involved in the catalytic synthesis of dopamine (map 00950) and dopaquinone (map 00350). Transcriptome profiling in August showed that 14 unigenes were aligned to EC 1.10.3.1, but no unigene was differentially expressed between upright and inverted *P. yunnanensis* cuttings. In spite of this finding, the greater number of upregulated genes (8) and higher sum of the log2FoldChange (13.74) observed under inversion than in upright conditions (6 and −8.01, respectively) may result in an accumulation of this activity (879.71 U·(μg·FW·min)^−1^ in inversion and 776.21 U·(μg·FW·min)^−1^ in upright), with significant changes at the 0.05 level, suggesting that PPO has a small effect on metabolism in inverted plants.

POD belongs to the antioxidant systems and plays a functional role in hydrogen peroxide detoxification and lignin biosynthesis [87,88]. A significant increase in POD activity is observed under abiotic stresses, such as drought [89] and salt stress [45]. According to IUBMB, this enzyme is identified as EC 1.11.1.7. It is also known as guaiacol peroxidase and plays an important role in lignin biosynthesis (Figure 7E). Lignin enhances plant cell wall rigidity, which is actively involved in the growth and development of trees and in the response to changing environmental conditions [90,91]. In this study, guaiacol peroxidase (E1.11.1.7) was examined by oxidizing guaiacol. Transcriptome analysis annotated thirteen DEGs to a peroxidase of lignin biosynthesis (map 00940), among which twelve unigenes were upregulated. This finding is in accordance with the analysis of POD activity, in which POD activities in inverted *P. yunnanensis* cuttings were always significantly higher than in upright cuttings. We therefore hypothesized that the cellular lignification process is accelerated to protect plants against oxidative stress under POD catalysis.

### 4.3. A Crucial Role of Cell Wall Metabolism in Response to Inversion of Plants is Revealed via RNA-Seq

What is the essential effect on the growth of inverted cuttings of *P. yunnanensis* when the biosynthesis and accumulation of hormones are unrelated to inversion? We compared the transcriptome profiles between upright and inverted samples. The KEGG results showed that most DEGs were enriched in carbohydrate metabolism, including amino sugar and nucleotide sugar metabolism (map 00520), pentose and glucuronate interconversions (map 00040), galactose metabolism (map 00052), and ascorbate and aldarate metabolism (map 00053). These four pathways included a total of 51 DEGs, among which 22 encoded 2 key enzymes: uridine diphosphate (UDP)-sugar pyrophosphorylase (USP, K12447, EC 2.7.7.64), encoded by 15 DEGs; and pectin methylesterase (PME, K01051, E3.1.1.11), encoded by 7 DEGs.

Cell wall polymer synthesis requires various nucleotide sugars, among which UDP-sugars are the main precursors to primary metabolites, storage compounds, structural components, glycoproteins, and glycolipids [92,93,94]. USP, with a broad substrate spectrum, can produce a variety of UDP-sugars and their analogs, which are essential for arabinose and xylose recycling during vegetative and reproductive growth in plants [95,96]. *Arabidopsis* USP (AtUSP) has no close homologs, and its disruption blocks normal male gametogenesis [97]. Overexpression of UDP-glucose pyrophosphorylase (UGP), which also contributes to the UDP-sugar form, can affect carbon allocation in hybrid *Populus*, resulting in a decrease in growth and an increase in defense metabolites [98]. In this study, the growth of inverted *P. yunnanensis* cuttings was inhibited, and USP-encoding transcripts were differentially expressed, with upregulation of 12 genes and downregulation of 3. Consequently, USP functions similarly to UGP, and changes in USP block the growth of plants.

Pectin is a major component of the cell wall and acts as a regulator of intercellular adhesion [99,100]. Pectin methylesterase (PME), also known as pectinesterase (PE), is an enzyme specific to pectin hydrolysis that removes the methoxy group from methylated pectin substances [101,102]. Changes in the activity of this enzyme accompany vegetative and reproductive processes of plants, including wood and pollen formation and plant-pathogen interactions [103,104,105,106]. The expression of fungal PME in transgenic tobacco affects cell wall metabolism and promotes the dwarfism phenotype [107]. The PME-encoding transcripts identified in this study were altered in inverted cuttings of *P. yunnanensis*, among which six unigenes were upregulated and one was downregulated. These changes lead to adverse pectin metabolism and a possible dwarf main branch.

## 5. Conclusions

Inverted cuttings of *P. yunnanensis* are characterized by a dwarf branch in response to inversion. Plant endogenous hormones including IAA and GA_3_, which are known as growth promoters, and ABA, known as a stress respondent, were, surprisingly, found to be unrelated to the inversion of the trees. Increasing POD activity in inverted plants serves as a promoter in lignin synthesis that functions in inverted tree cuttings. Additionally, changes in carbohydrate metabolism impact vegetative plant growth. Furthermore, the altered activities of USP for nucleotide sugars, PME for pectin and POD for lignin, are important factors that respond to the inversion of trees, and these activities are all involved in cell wall metabolism. We conjecture that the cell wall plays a crucial role during the vegetative growth of inverted trees. However, the involvement of the cell wall in response to the inversion of plants requires further study.

## Figures and Tables

**Figure 1 genes-09-00572-f001:**
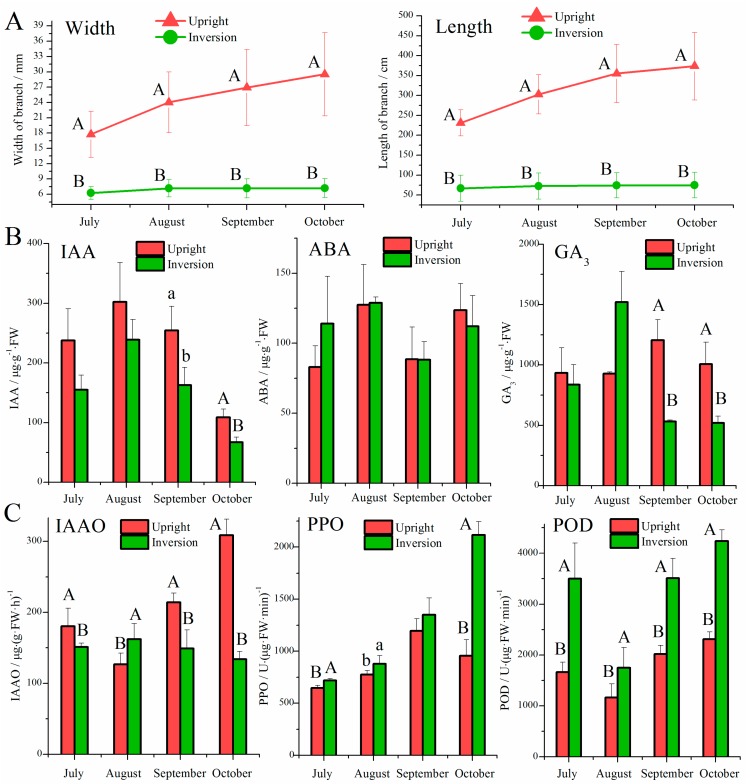
Comparison of vegetal and physiological characteristics between upright and inverted cuttings. (**A**) Width and length of branches. (**B**) Contents of indole-3-acetic acid (IAA), abscisic acid (ABA) and gibberellic acid (GA_3_) in the top leaves. (**C**) Activities of IAA oxidase (IAAO), polyphenol oxidase (PPO) and peroxidases (POD) in the top leaves. Data are given as the mean ± standard error (SE). Different lowercase letters and capital letters indicate significant differences between upright and inverted cuttings at the 0.05 and 0.01 levels, respectively.

**Figure 2 genes-09-00572-f002:**
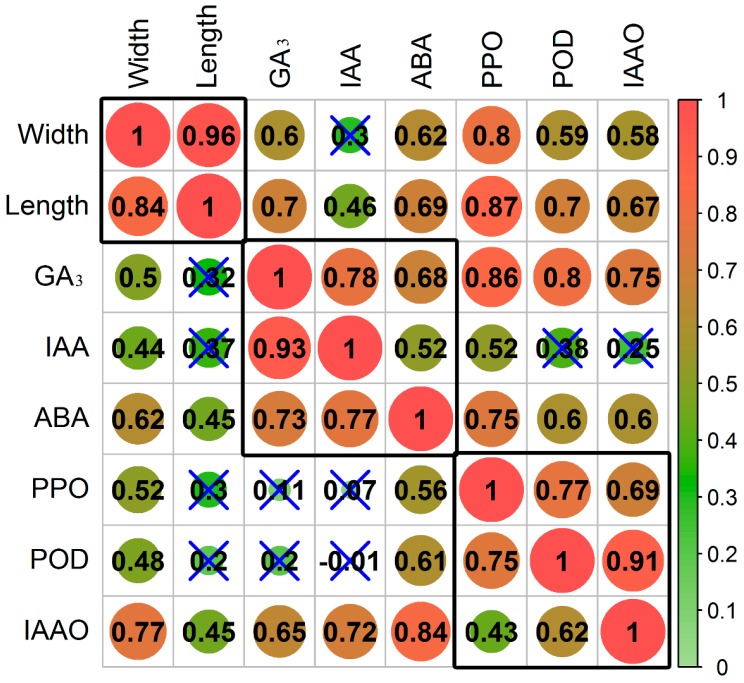
Correlations between growth, hormones, and enzymes. The left diagonal shows the correlations of inverted cuttings, and the right diagonal shows the correlations of upright cuttings. The correlation coefficient without significance is marked with a blue “×” symbol.

**Figure 3 genes-09-00572-f003:**
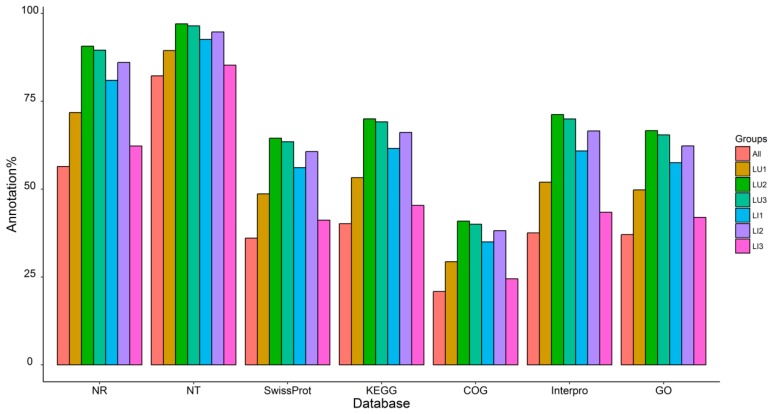
Functional annotation of unigenes. NR: non-redundant protein sequences; NT: non-redundant nucleotide sequences; KEGG: Kyoto encyclopedia of genes and genomes; COG: clusters of orthologous groups of proteins; GO: gene ontology.

**Figure 4 genes-09-00572-f004:**
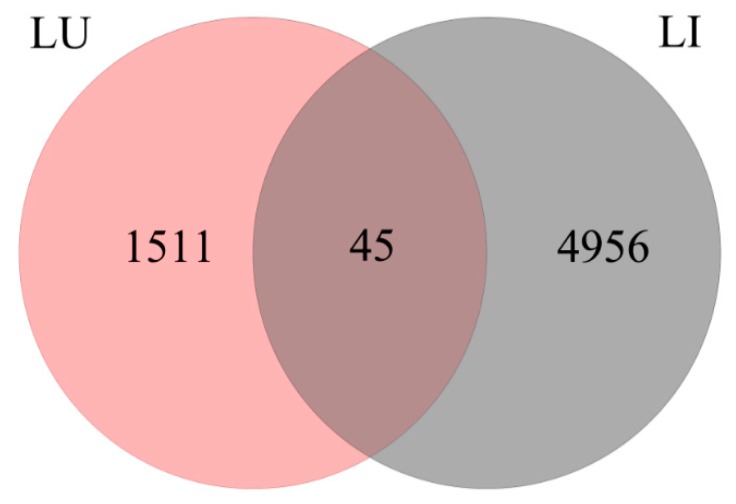
Venn diagram of differentially expressed genes (DEGs) identified through pairwise comparison between group LU and group LI.

**Figure 5 genes-09-00572-f005:**
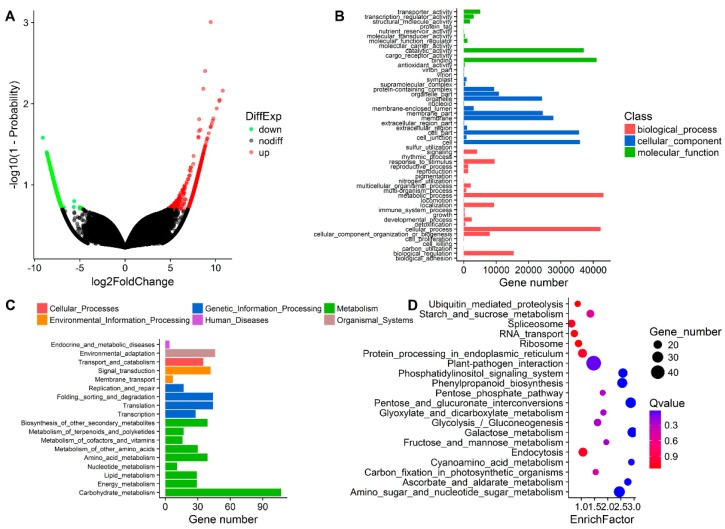
Functional enrichment of differential gene expression profiling in LU and LI. (**A**) Volcanic plot showing differential expression levels. (**B**) GO enrichment of DEGs. (**C**) KEGG pathway levels of DEGs. (**D**) KEGG enrichment of DEGs.

**Figure 6 genes-09-00572-f006:**
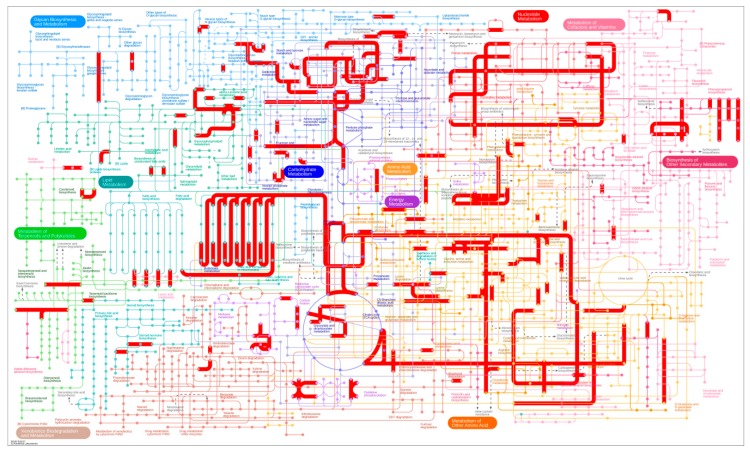
An iPath metabolic map.

**Figure 7 genes-09-00572-f007:**
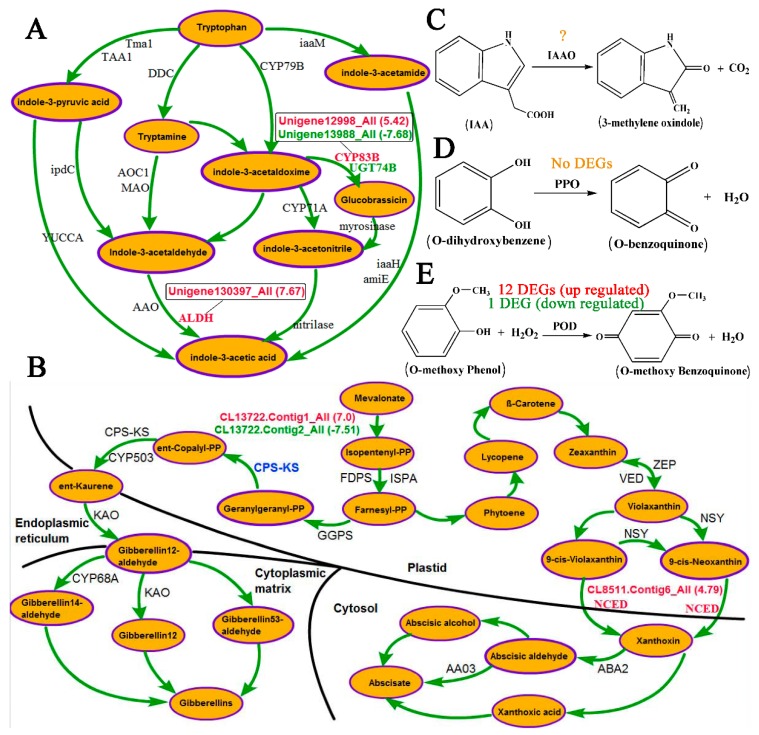
Effects of inversion on the IAA, ABA, and GA pathways and the catalytic reactions of IAAO, PPO, and POD. (**A**) The biosynthesis of IAA. (**B**) The biosynthesis of ABA and GA. (**C**) The reaction catalyzed by IAAO. (**D**) The reaction catalyzed by PPO. (**E**) The reaction catalyzed by POD. Red font indicates an upregulated unigene(s) or enzyme(s). Green font indicates a downregulated unigene(s) or enzyme(s). Blue font indicates the enzymes associated with both upregulated and downregulated unigenes. Yellow font indicates the enzymes with no unigenes.

**Table 1 genes-09-00572-t001:** Statistics of sequenced data and assembly results. Total RNA was analyzed from the top leaf samples of upright (LU) and inverted (LI) cuttings of *P. yunnanensis.*

Samples	Reads				Unigenes			
	Total Raw Reads (Mb)	Total Clean Reads (Mb)	Clean Reads Q20 (%)	Clean Reads Q30 (%)	Total Number	Mean Length (nt)	N50	N90
LU1	64.78	44.75	98.09	94.47	127876	543	826	238
LU2	64.78	44.39	98	94.19	68747	805	1336	310
LU3	63.16	44.8	98.05	94.32	73913	857	1471	323
LI1	63.16	44.86	98.03	94.27	100268	741	1354	272
LI2	64.78	45.17	97.94	94.03	86133	784	1386	285
LI3	64.78	44.31	97.86	93.9	210767	560	801	246
All	-	-	-	-	296815	693	1017	294

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
