# Peer review of "Physiological Analysis and Transcriptome Profiling of Inverted Cuttings of Populus yunnanensis Reveal That Cell Wall Metabolism Plays a Crucial Role in Responding to Inversion"

_genes, 2018, doi:10.3390/genes9120572_

Round 1

Reviewer 1 Report

Review for Genes (Zhou et al. Physiological analysis and …polarity stress)

The manuscript by Zhou and co-workers attempts to show the changes in plant hormones and transcriptome in response to the inverted planting of Populus shoot cuttings. The study goes to identify the changes in hormonal abundances and transcriptional changes under upright and inverted shoot plantings. It appears that the experiments in this study are vividly performed to establish the link between hormones and metabolism.  The authors take some well-established, complementary approaches, including gene expression analyses by qRT-PCR, RNA-seq, and hormone measurements by HPLC. 

The article adds to the knowledge in areas of plant propagation by establishing links between hormones and vegetative propagation. The strength of this study is revealing the changes in the expressions of various genes involved in hormones, and metabolisms associated with sugars, lipids, carotenoids, cell-wall components. Nevertheless, I have a few comments.

 1.      The authors may need to clarify the context of the term “polarity” used in this manuscript. I assume that “Plant polarity (Zygotes/spores) ” is different from “Polarity in auxin transport (inverted plants??)”. The introduction to the manuscript must define the terminology. Furthermore, the authors introduces the term ‘totipotency’- the feature unrelated to the context of inverted planting. The introduction section may need a through revision to clarity. 

2.      Any undue changes in the physiological process are stressful to plants. The authors claim that the inversion planting induces polarity stress similar to abiotic stress”- this needs to be revised. What are the abiotic factors inducing stress when plants are inverted needs to be stated clearly? Inversions of shoot apices are common to some plant growth habits. For instance, in plants those are runners (eg. Strawberry) it is natural that shoots are inverted when the apices bends to reach the soil and rooting occurs. This is also a type of layering propagation. Here the physiological changes due to bending and other resource signals (from soil in place of light) alter the identity of few cells to take up root development programs. Essentially, inverted planting of cuttings would be altering the direction of signals derived from the shoot apex and basal part. 

3.    If possible, the authors should provide a diagram or photograph showing the two treatments-upright and inverted positions of the shoot cuttings. What was the size (length and diameter) of the cuttings used in the treatments? Please bring in more clarity to the methodology. Describe how the cuttings were processed for culturing, soil type, temperature, seasonal changes at the time of the experiment, time – the interval between the planting of cuttings and sprouts samplings, watering, prophylactic procedures taken prevent disease/pest incidences. 

4.    The sampling procedure should be explained to bring in clarity. A portion of the cutting or leaves formed from sprouts or entire sprout (shoots+leaves) were harvested for hormone quantification? Which tissues were used for RNA-seq experiments- sprouts or leaves or cuttings? What was the size of the tissues sampled?  At what developmental stage of bud burst were the tissues collected?  Per Line 136-  six samples were collected- what tissues were these samples? How many from samples per direction type were harvested? These details are relevant to the interpretation of the findings. 

5.    The RNA-seq results pertaining to the significance of differential expressions of genes should be represented as Venn-diagrams for easy interpretation of the findings. This diagram should provide break-down of common and unique differentially expressed gene categories. 

6. Line 146-change ‘ware’ to ‘were’

Author Response

Thank you for your suggestion. We have carefully revised the text of the manuscript according to your comments. Our responses to the comments are uploaed as a Word file

Reviewer 2 Report

In this manuscript, authors did physiological analysis and transcriptome profiling of inverted cuttings of Populus yunnanensis that reveal cell wall metabolism has a crucial role in responding to polarity stress. The manuscript is very well written. However, for the betterment of this manuscript, I have few suggestions.

Major:-

1.    It would be nice if the authors showed some physiological changes as represented by morphological images of Populus.

2.    In introduction add one paragraph about all the previous genome-wide profiling and abiotic stress studies such as

a. Genome-wide analysis of gene expression profiling revealed that COP9 signalosome is essential for correct expression of Fe homeostasis genes in Arabidopsis.

b. Genome-Wide Identification and Analysis of Genes, Conserved between japonica and indica Rice Cultivars, that Respond to Low-Temperature Stress at the Vegetative Growth Stage. Front. Plant Sci. 8:1120. doi: 10.3389/fpls.2017.01120.

c. Genome-Wide Analysis of the PYL Gene Family and Identification of PYL Genes That Respond to Abiotic Stress in Brassica napus.

d. Ectopic expression of OsSta2 enhances salt stress tolerance in rice. Front. Plant Sci. 8:316. doi: 10.3389/fpls.2017.00316.

Minor:

L45-L49 Rephrase this line.

Author Response

(The authors gave the same response as above.)

Round 2

Reviewer 1 Report

Thank you for revising the manuscript. I don't have any major comments on this version except that the new version needs to be checked throughly for english grammar and typological errors. 

Reviewer 2 Report

I am happy with the modification. This MS can be accepted.